# Chiral phosphoric acid-catalyzed stereodivergent synthesis of trisubstituted allenes and computational mechanistic studies

Jiawen Wang[1,2,4], Sujuan Zheng[3,4], Subramani Rajkumar[1], Jinglei Xie[1], Na Yu[1], Qian Peng [3✉] &
Xiaoyu Yang [1✉]

Chiral molecules with multiple stereocenters are widely present in natural products and pharmaceuticals, whose absolute and relative configurations are both critically important for their physiological activities. In spite of the fact that a series of ingenious strategies have been developed for asymmetric diastereodivergent catalysis, most of these methods are limited to the divergent construction of point chirality. Here we report an enantioselective and diastereodivergent synthesis of trisubstituted allenes by asymmetric additions of oxazolones to activated 1,3-enynes enabled by chiral phosphoric acid (CPA) catalysis, where the divergence of the allenic axial stereogenicity is realized by modifications of CPA catalysts. Density functional theory (DFT) calculations are performed to elucidate the origin of diastereodivergence by the stacking- and stagger-form in the transition state (TS) of allene formation step, as well as to disclose a Münchnone-type activation mode of oxazolones under Brønsted acid catalysis.

[1] School of Physical Science and Technology, ShanghaiTech University, Shanghai 201210, China. [2] University of Chinese Academy of Sciences, 100049 Beijing, China. [3] State Key Laboratory of Elemento-Organic Chemistry, College of Chemistry, Nankai University, 300071 Tianjin, China. [4] These authors contributed equally: Jiawen Wang, Sujuan Zheng. ✉email: qpeng@nankai.edu.cn; yangxy1@shanghaitech.edu.cn

Chiral allenes are featured in many biologically active natural products, pharmaceuticals, and functional materials[1,2]. In addition, they also serve as versatile building blocks in organic synthesis due to their diverse reactivities[3]. Despite of the high demands of chiral allenes, the asymmetric catalytic synthesis of these axially chiral compounds remains a challenge in organic synthesis[4–6]. In the last two decades, a number of elegant asymmetric catalytic strategies have been developed for chiral allene synthesis, such as nucleophilic additions of 1,3-enynes[7–19], dynamic kinetic asymmetric transformations (DyKAT) of racemic allenes[20–22], rearrangement of alkynes[23–25], coupling of alkynes with diazo compounds[26–28] and others[29–31]. Among these strategies, the direct asymmetric additions of prochiral 1,3-enynes represent as one of the most attractive strategies for synthesis of multiple-substituted chiral allenes, owing to the easy accessibility of these substrates. Since the pioneer work of enantioselective synthesis of boryl, silyl and aryl allenes via chiral Pd and Rh catalyzed asymmetric additions of 1,3-enynes by Hayashi and co-workers[7–9], a series of elegant asymmetric reactions have been developed employing this strategy, either through asymmetric transition metal-catalysis[7–14] or organocatalysis[15–19] (Fig. 1a).

Chiral molecules with multiple stereocenters are widely present in natural products and pharmaceuticals, whose absolute and relative configurations are both critically important for their physiological activities. In the past few decades, numerous highly enantioselective and diastereoselective reactions have been developed. However, modulation the sense of diastereoselectivity in an asymmetric catalytic reaction is still challenging, because the diastereochemical preference is largely governed by the inherent structure and stereoelectronic nature of the substrates[32]. To address this intrinsic problem, a series of ingenious strategies have been developed for asymmetric diastereodivergent catalysis[33–35], such as using distinct catalysts[36–38], change of metal cations[39], and ligands[40,41] of the catalysts, change of reaction conditions[42,43], stereodivergent dual catalysis[44–50] and stepwise control[51–54]. Nevertheless, achieving asymmetric diastereodivergent catalysis through modifications of one single type of chiral catalysts remains elusive[55,56].

Despite the fact that a large number of asymmetric stereodivergent catalytic methods have been developed, most of these methods have been limited to the construction of point stereogenicity divergence, while stereodivergent synthesis of axial stereogenicity has been rarely explored, except using stepwise

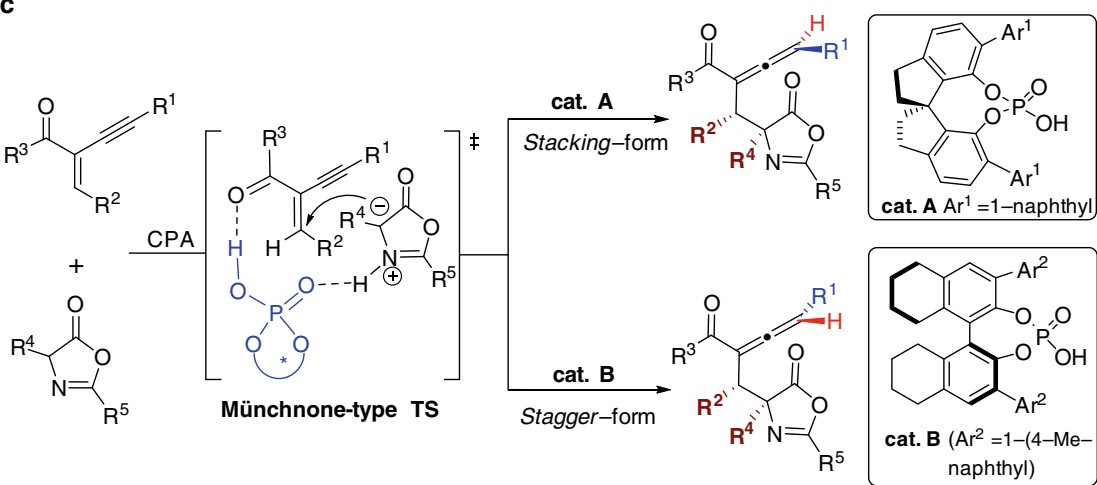

**Fig. 1 Asymmetric synthesis of chiral allenes and stereodivergent synthesis. a** Asymmetric synthesis of chiral allenes via enantioselective additions of activated 1,3-enynes. **b** Asymmetric stereodivergent catalysis was limited to the construction of point stereogenicity divergence. **c** Asymmetric stereodivergent construction of axial stereogenicity via modifications of chiral phosphoric acid catalysts.

control strategy[53,54] (Fig. 1b). Herein, we report an enantioselective and diastereodivergent synthesis of trisubstituted allenes via asymmetric conjugate additions of activated 1,3-enynes by oxazolones[57–60] enabled by CPA catalysis, in which the diastereodivergent construction of the allenic axial chirality is realized by modifications of CPA catalysts (Fig. 1c). In addition, the origin of the diastereodivergence is well elucidated by DFT calculations, in which a Münchnone-type activation mode of oxazolones under Brønsted acid catalysis is presented.

## Results

**Reaction optimizations.** We commenced our study by selecting α-alkynyl-α,β-enone **1a** and 2-*para*-methoxyphenyl (PMP) substituted oxazolone **2a** as model substrates under CPA catalysis (Table 1). Under the promotion of TRIP catalyst (CPA **A1**, 10 mol%) in toluene (with 3 Å molecular sieves) at room temperature, only two diastereomeric allene products among the four potential ones were detected (Table 1, entry 1), albeit with both poor diastereomeric ratio (dr, **3a**:**4a** 1:2.8) and enantiomeric excess (ee). Subsequently, a series of BINOL and H8-BINOL derived CPA catalysts were screened (entries 2–7). Satisfyingly, the 1-(4-Me-naphthyl) substituted H8-BINOL-derived catalyst **B2** provided the allene product in 91% yield with both high diastereoselectivity and enantioselectivity (**3a**:**4a** 10:1, 87% ee, entry 7). However, surprisingly, switching the chiral scaffold of

1-naphthyl substituted CPA catalyst form H8-BINOL-type to SPINOL-type (CPA **C1**) led to the reversal of diastereoselectivity (**3a**:**4a** 1:11, entry 8) and formation of product **4a** with high ee as well, albeit with moderate yield. To obtain better stereoselectivity control and improve the yield, the effect of the R group at the 2-position of oxazolone was exploited (see Supplementary Table 1), which indicated that 3,5-dimethoxyphenyl group was the optimal one (entry 9 and 10). Finally, a variety of solvents were also screened, and CCl₄ was chosen as the optimal solvent (see Supplementary Table 1 for details), in which the chiral allene **3a** was obtained in 98% yield, 20:1 dr (**3a**:**4a**) with 91% ee in the presence of CPA **B2**, while diastereomeric chiral allene **4a** was generated in 85% yield, 12:1 dr (**4a**:**3a**) with 98% ee under the catalysis of CPA **C1** (entries 11–12).

**Substrate scope.** Having established the optimal conditions for stereodivergent synthesis of chiral trisubstituted allenes via modifications of CPA catalysts, the substrate scope under the catalysis of CPA (*S*)-**B2** was firstly investigated (Fig. 2). Various substituted phenylacetylenyl groups were well tolerated under the optimal conditions, regardless of the electronic nature and positions of the substitutions, affording the allene products **3a**–**3h** with high diastereoselectivities (>7:1) and enantioselectivities. In addition, substitutions of the R¹ group with heteroaryl, alkenyl and alkyl groups were also amenable, which yielded the products

**Table 1 Optimizations of reaction conditions[a].**

| Entry | R | Catalyst | Solvents | Yield[b] (%) | dr[b] (3a:4a) | ee[c] (3a/4a, %) |
|---|---|---|---|---|---|---|
| 1 | PMP | **A1** | toluene | 52 | 1:2.8 | 45/31 |
| 2 | PMP | **A2** | toluene | 41 | 1:1 | 41/24 |
| 3 | PMP | **A3** | toluene | 51 | 1.4:1 | 35/57 |
| 4 | PMP | **A4** | toluene | 50 | 1.3:1 | 72/81 |
| 5 | PMP | **A5** | toluene | 68 | 1:2.5 | 24/90 |
| 6 | PMP | **B1** | toluene | 57 | 2.7:1 | 91/34 |
| 7 | PMP | **B2** | toluene | 91 | 10:1 | 87/– |
| 8 | PMP | **C1** | toluene | 51 | 1:11 | –/98 |
| 9 | Ar | **B2** | toluene | 99 | 12:1 | 91/– |
| 10 | Ar | **C1** | toluene | 80 | 1:9 | –/94 |
| 11 | Ar | **B2** | CCl₄ | 98 | 20:1 | 91/– |
| 12 | Ar | **C1** | CCl₄ | 85 | 1:12 | –/98 |

[a]Reactions were performed with **1a** (0.15 mmol), **2a** (0.1 mmol), cat (0.01 mmol), 3 Å MS (100 mg), solvents (0.5 mL) at ambient temperature for 24 h.
[b]Yields and dr value were determined by crude ¹H NMR analysis using 1,2-dimethoxyethane (DME, 0.1 mmol) as internal standard.
[c]ee values were determined by HPLC analysis on a chiral stationary phase. PMP = *para*-methoxylphenyl, **Ar** = 3,5-dimethoxyphenyl.

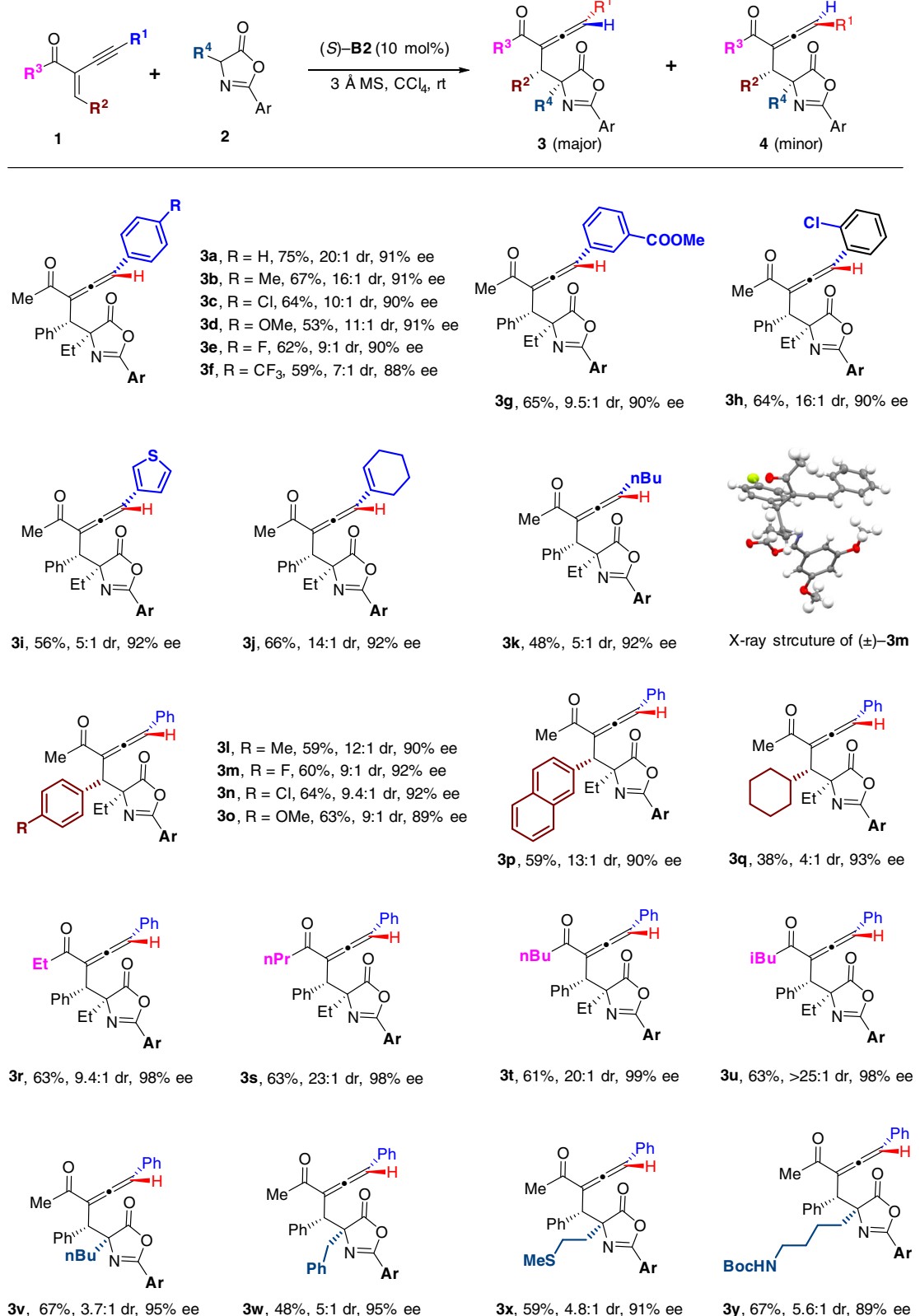

**Fig. 2 Scope for asymmetric synthesis of trisubstituted allenes 3 catalyzed by CPA catalyst (S)-B2.** Unless otherwise noted, reactions were performed with **1** (0.15 mmol), **2** (0.1 mmol), (S)-**B2** catalyst (0.01 mmol) and 3 Å MS (100 mg) in CCl$_4$ (0.5 mL) at room temperature for 24 h. Yields were isolated yields of allenes **3**. Dr values were determined by crude $^1$H NMR analysis. Ee values were determined by HPLC analysis on a chiral stationary phase. **Ar** = 3,5-dimethoxyphenyl.

with high enantioselectivities, albeit with moderate to high diastereoselectivity control (**3i–3k**). Next, a range of R$^2$ groups at the β-positions of the enones were explored, which suggested that various substituted aryl groups (**3l–3p**) were well tolerated, as well as an alkyl group, albeit with moderate dr value (**3q**). The relative configurations of the allene products **3** were assigned by analogy to **3m**, whose relative structure was confirmed by X-ray crystallography. Subsequently, a range of groups at the ketone site (R$^3$) were also investigated, where the Et-substituted, *n*Pr-substituted, *n*Bu-substituted, and *i*Bu-substituted substrates all generated the chiral allenes with both excellent enantioselectivitie and diastereoselectivitie (>9.4:1 dr, >98% ee, **3r–3u**). Finally, a series of substitutions (including some functional group-containing substituents) at the 4-position of oxazolones were also exploited

under the optimal conditions, which indicated that the chiral allene products could be produced with high enantioselectivities, albeit with a bit erosive diastereoselectivities (3.7:1–5.6:1, **3v–3y**).

After investigation of the scope under (*S*)-**B2** catalysis, the substrate scope generality with catalyst (*R*)-**C1** was also studied (Fig. 3). All the R$^1$, R$^2$ and R$^4$ substituted substrates explored in Fig. 2 were subjected into investigation under the catalysis of (*R*)-**C1** catalyst, which generated the diastereomeric allenes **4** as the major products with both high dr and ee values for most cases (**4a–4y**). The absolute structures of the chiral allenes **4** were assigned by analogy to product **4c**, whose absolute configuration was unambiguously confirmed by X-ray crystallography. Unfortunately, the variations of the R$^3$ groups of α-alkynyl-α,β-enone **1** under standard conditions were not well tolerated, which led to

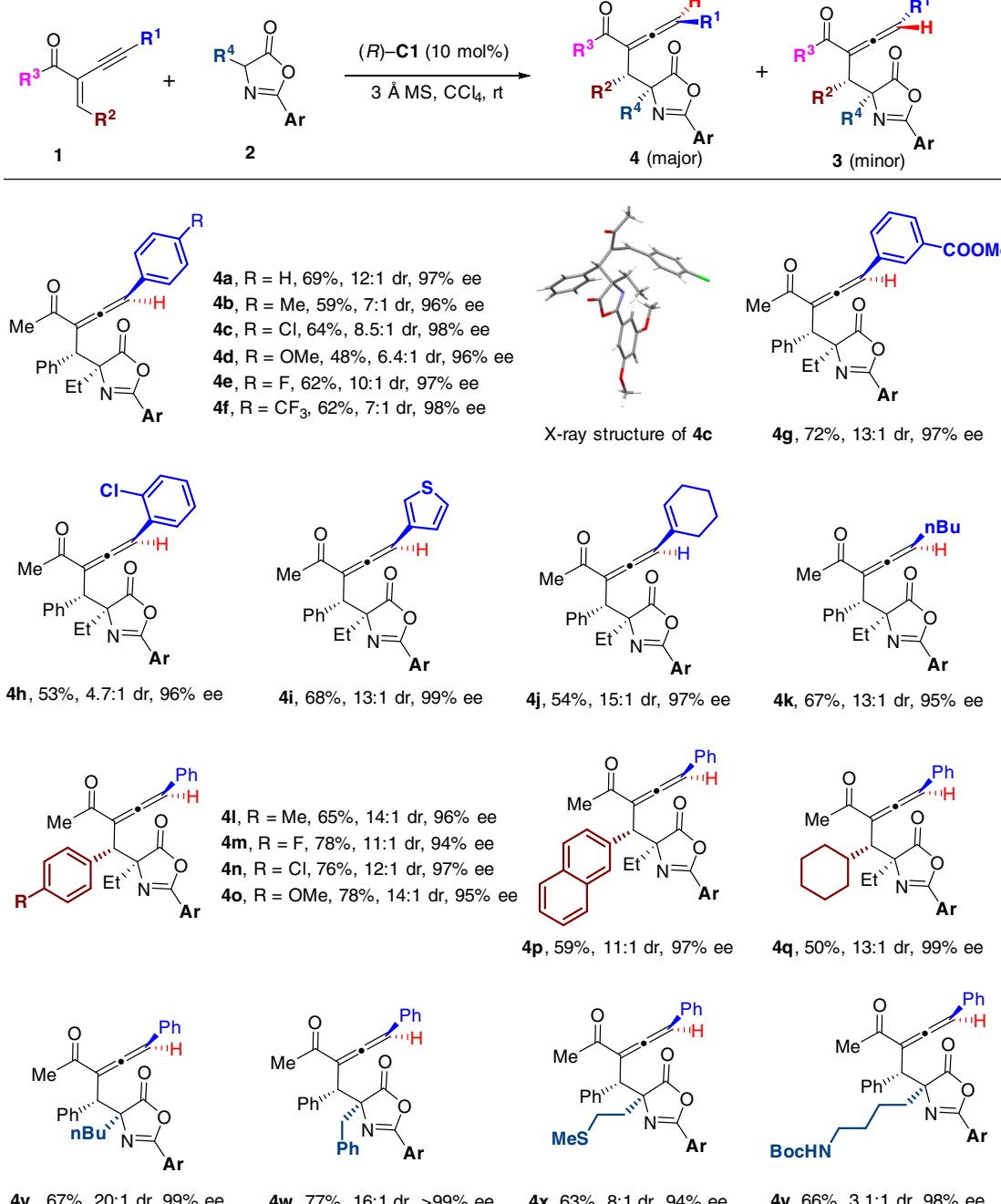

**Fig. 3 Scope for asymmetric synthesis of trisubstituted allenes 4 catalyzed by CPA catalyst (R)-C1.** Unless otherwise noted, the reaction conditions were the same with those indicated in Fig. 2, except (*R*)-**C1** catalyst (10 mol%) was used instead. **Ar** = 3,5-dimethoxyphenyl.

very low yields and decreased diastereoselectivities (see Supplementary Fig. 1 for details).

**Reaction mechanism and origin of diastereodivergence.** To investigate the origin of diastereodivergence in construction of the allenic axial chirality, a control experiment was performed. In the presence of CPA catalysts (S)-**B2** and (R)-**C1**, respectively, the asymmetric isomerization of racemic α-alkynyl ketone[23] **5a** proceeded efficiently to give the chiral trisubstituted allene **6a** in the opposite enantiomeric bias (Fig. 4a). In addition, some control experiments of the stabilities of the chiral allene products under various conditions were performed (see Supplementary Fig. 2), which indicated that the conversion from a kinetic epimer to a thermodynamic epimer under the reaction conditions is probably not likely. Based on these results and previous reports[16–18], a preliminary stepwise mechanism was proposed: (1) oxazolones were activated by CPA catalyst via tautomerism to form active enol intermediates; (2) the Michael addition of α-alkynyl enones by the activated oxazolone intermediates generated another enol intermediates **INT A** stereoselectively under the guidance of CPA catalysts; (3) the CPA catalyst mediated the proton transfer[61,62] of **INT A** to generate products with the allenic axial stereogenicity (Fig. 4b). For the stereoselectivities of these reactions, we presumed that both (S)-H8-BINOL-**B2** and (R)-SPINOL-**C1** catalyst generated the same (S,R)-syn-configuration in the Michael addition step, which was followed by the proton transfer step to afford the distinct axial chirality of (S,R,R)-product or (S,R,S)-product, respectively.

To unveil the different stereochemical control of CPA catalysts in Michael addition and proton transfer steps, density functional theory (DFT) calculations were performed at M06-2X/6-311++G(d,p)//M06-2X//6-31G(d) level of theory[63–65] (see Supplementary Data 1 for Cartesian coordinates of the optimized structures). Surprisingly, based on a well-accepted enol-type mode (mode A) of oxazolones[57,58], initial calculations of the Michael addition step using an achiral phosphoric acid catalyst (dimethyl phosphate, **DMP**) provided the anti-addition predication of stereochemical outcomes, which were not in agreement with the experimental syn-addition results (Fig. 4c). Further computational evaluations using chiral (S)-**B2** and (R)-**C1** catalysts remain the wrong (R,R)-anti-addition prediction as well, suggesting that the diastereoselectivity of the Michael addition step did not rely on the chiral scaffolds of CPA catalysts (see Supplementary Fig. 4). Inspired by the basicity of related functional groups of oxazolone, the imine part of oxazolones acting as proton accepter in hydrogen bonding with CPA catalyst is more accessible rather than the carbonyl group (see Supplementary Fig. 4a). Therefore, a Münchnone-type mode[59,60,66] (**mode B**) was proposed and the correct syn-addition outcomes were achieved depended on the achiral **DMP** catalyst (Fig. 4c). The calculation results suggested that transition state (TS) of enol-type requires 6.1 kcal/mol more activation energy than that of the Münchnone-type by achiral **DMP** catalyst in the prototropic activation step. For the Michael addition step, Münchnone-type model is also superior to enol-type one. And among a number of our calculated Newman conformations (see Supplementary Table 2), the most favored conformation of syn-addition-TS in Münchnone-type is more stable than the most favored conformation of anti-addition-TS in enol-type by 4.2 kcal/mol, indicating this diastereoselectivity is highly model dependent. The calculations, based on chiral (S)-**B2** and (R)-**C1** catalysts, provided similar results that phosphoric acid catalysts tend to activate oxazolones in a fashion of Münchnone-type mechanism in these reactions, leading to syn-addition products for the diastereoselectivities (see Supplementary Figs. 3 and 4).

The energy profile of Münchnone-type mechanism was performed in Fig. 4d. Because only the Michael addition and proton transfer steps are responsible for the chirality control in these reactions, the prototropic activation steps forming the key precursor **INT2** were not shown here (for the details of these steps see Supplementary Fig. 3). From the related **INT2-C/-B**, the activated 1,3-enyne **1a** undergoes a nucleophilic attack by Münchnone-type intermediate of oxazolone via **TS2** accompanying the proton delivery of CPA catalyst. There are four major transition states with different chiral features located, namely syn-isomers (**TS2-RS**, **TS2-SR**), anti-isomers (**TS2-RR**, **TS2-SS**). Although the CPA catalysts of (S)-**B2** and (R)-**C1** provide quite similar chiral cavities, they surprisingly display distinct modes of stereoselectivity control in both Michael-addition and proton transfer steps depended on the flexible BINOL backbone and rigid SPINOL backbone, respectively. For (S)-H8-BINOL-**B2** catalyst, the final configuration of product **Pro-SRR** is determined by the proton transfer step with 1.7 kcal/mol free energy difference after the facile and reversible Michael-addition step. Due to sterically repulsive interactions between the phenyl group (R²) at the β-position of **1a** and the ethyl group of **2a**, the **TS2-B-RS** is unstable for 1.9 kcal/mol comparing to the favorable **TS2-B-SR** that is supported by the distortion interaction analysis in Supplementary Fig. 5b. By using the (R)-SPINOL-**C1** catalyst, the Michael-addition step mainly determined the reaction rate and led to the more stable **INT3-C-SR** intermediate readily for proton transfer step forming **Pro-SRS**. From Supplementary Fig. 5b, the interaction energy of (R)-**C1** catalyst and substrate dominates the energy difference of **TS2-C-SR** and **TS2-C-RS** probably because of the steric repulsion between 3,3'-substitutent of the **C1** catalyst and the ethyl group of **2a**, indicating the rigid conformation of SPINOL-**C1** catalyst may encounter non-negligible interactions between catalyst and substrate, compared with the relatively flexible conformation of H8-BINOL-**B2** catalyst. The stereoisomerization of enol intermediate **INT 3** via low barrier tautomerization and single bond rotation was omitted for clarity. Finally, our calculation predicted the products with 95% ee, 17:1 dr (exp. 91% ee, 20:1 dr) and 98% ee, 15:1 dr (exp. 97% ee, 12:1 dr) under (S)-**B2** and (R)-**C1** catalysis, respectively, which are in agreement with the experimental results.

For the origin of chirality control in the proton transfer step for construction of the allenic axial chirality, we found two types of models among our 20 calculated TSs structures (see Supplementary Table 3). In Fig. 5a, stacking-type model tends to form S-axial chirality configuration TSs, where the substitutions of the intermediate in transition state **TS3** have strong intramolecular interaction (i.e., π-π interaction supported by the NCIs plots[67] of Fig. 5c) and the resonance stabilization between the phenyl group (R¹) and the allenic moiety in **1a** (see Supplementary Fig. 6 for details). In contrast, stagger-type model prefer R-axial chirality configuration TS with emphasizing the intermolecular interactions of CPA catalyst and substrates (for the distortion-interaction analysis and NCIs plots of TSs see Supplementary Figs. 6 and 7). Due to the relatively rigid conformation of (R)-SPINOL-**C1** catalyst, stacking-type model is favored for the TSs of proton transfer step to fit the cavity of catalyst with certain entropy loss, leading to S-axial chirality configuration. That is consistent with less steric repulsion between catalyst and substrate in the stacking-type model, which is the dominant effect by Energy Decomposition Analysis (EDA) calculations[68] as shown in Fig. 5b. The (S)-H8-BINOL-**B2** catalyst with slightly flexible conformation are more likely to adopt stagger-type mode forming R-axial chirality configuration, which is mainly stabilized by the dispersion effect by EDA calculation. As shown in NCIs plots in Fig. 5c, this dispersion effect between the methyl group of catalyst **B2** and substrate in **TS3-B-SRR** can well rationalize the catalyst

**a** **Control experiment:**

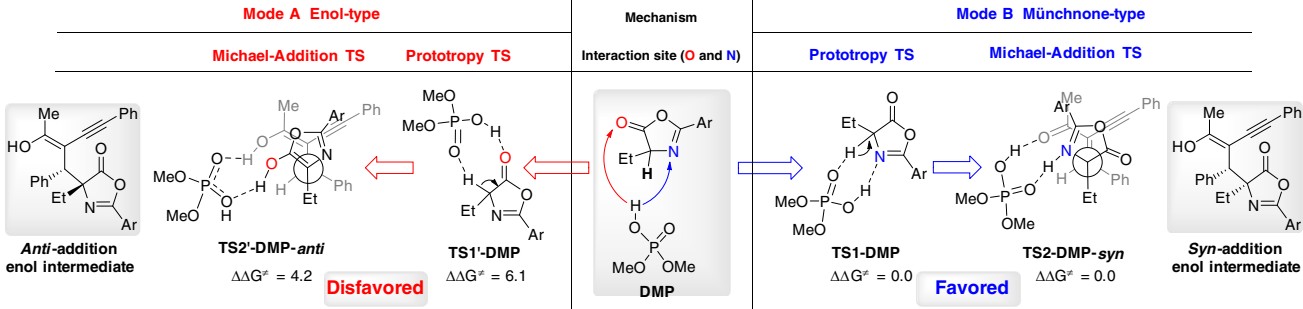

**b** **Initially proposed mechanism:**

**c** **Activation modes (Prototropy and Michael-Addition):**

| Mode A Enol-type | | Mechanism | Mode B Münchnone-type | |
|---|---|---|---|---|
| Michael-Addition TS | Prototropy TS | Interaction site (O and N) | Prototropy TS | Michael-Addition TS |

*Anti*-addition enol intermediate TS2'-DMP-*anti* ΔΔG‡ = 4.2 **Disfavored** TS1'-DMP ΔΔG‡ = 6.1 DMP TS1-DMP ΔΔG‡ = 0.0 TS2-DMP-*syn* ΔΔG‡ = 0.0 **Favored** *Syn*-addition enol intermediate

**d** **Energy profiles of Michael-Addition and Proton transfer with chiral catalysts:**

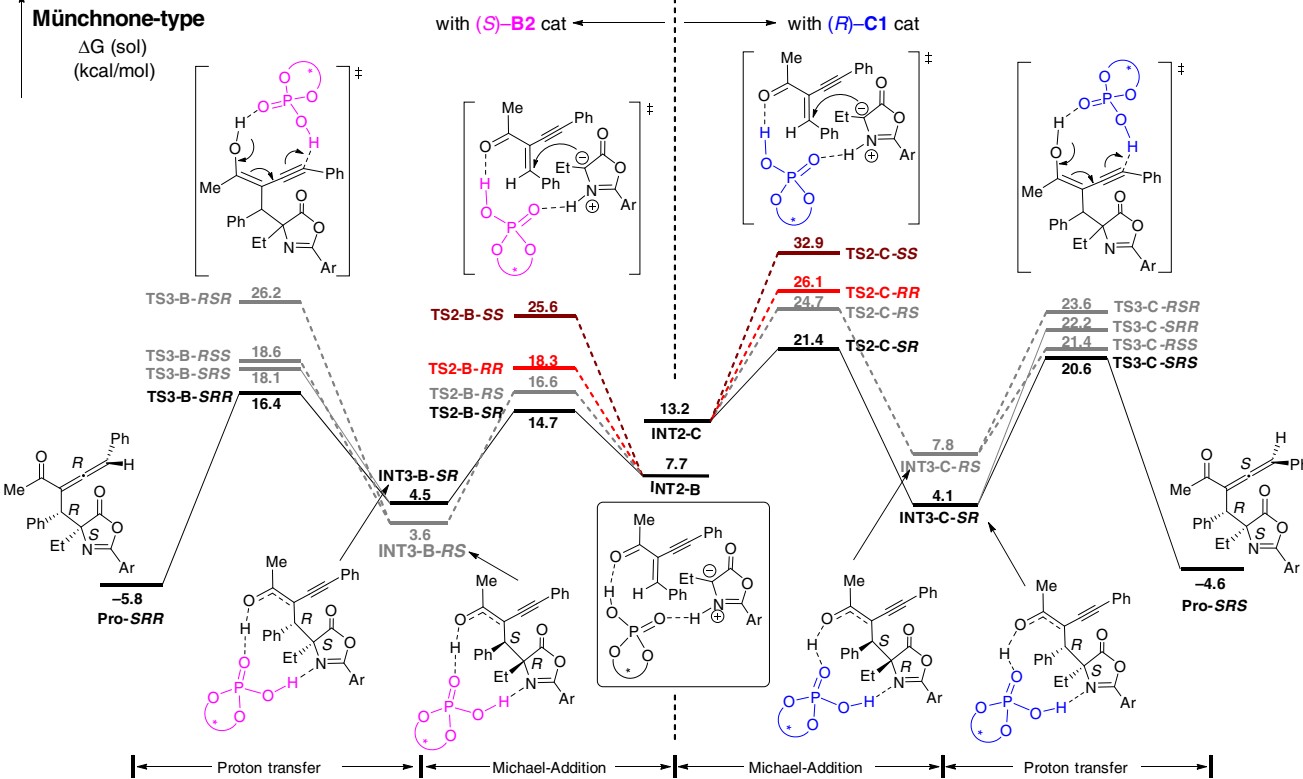

**Fig. 4 Mechanistic studies. a** Control experiments. **b** Initially proposed enol-type mode mechanism. **c** Comparison of the enol-type and Münchnone-type activation modes via calculation. **d** Energy profiles of Michael-Addition and proton transfer with CPA catalysts via DFT calculations. (The energy shown in kcal/mol, Ar = 3,5-dimethoxyphenyl).

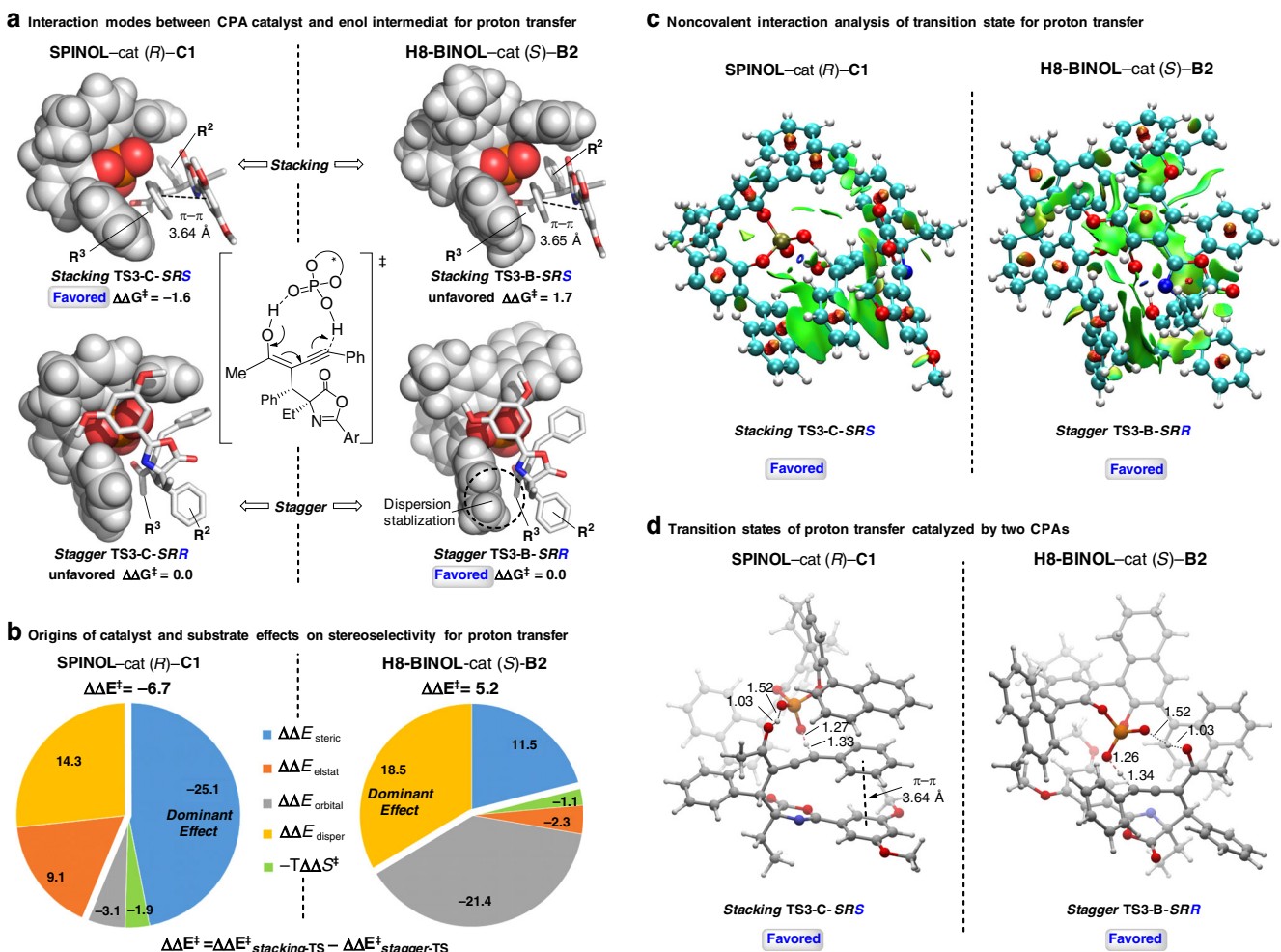

**Fig. 5 Computational studies on origin of diastereoselectivity.** Comparison two types of interaction modes between CPA (*R*)-**C1**/(*S*)-**B2** and substrates. **a** Space-filling model of *stacking*-TS versus *stagger*-TS for proton transfer. **b** Energy Decomposition Analysis (EDA) of transition states for proton transfer at B3LYP-D3/TZ2P level of theory by using ETS-NOCV in ADF. The relative energy ($\Delta\Delta E^{\ddagger}$) is the sum of $\Delta\Delta E_0^{\ddagger}$ and ($-T\Delta\Delta S^{\ddagger}$), where the entropy term is from Gaussian calculations. For the energy decomposition analysis equation follows: $\Delta\Delta E_0^{\ddagger} = \Delta\Delta E_{steric} + \Delta\Delta E_{elstat} + \Delta\Delta E_{orbital} + \Delta\Delta E_{disper}$. The $\Delta\Delta E^{\ddagger}$ is calculated from the electronic energy difference between *stacking*-type TS and *stagger*-type TS ($\Delta\Delta E^{\ddagger} = \Delta E_{stacking\text{-}TS}^{\ddagger} - \Delta E_{stagger\text{-}TS}^{\ddagger}$). Each energy component ($\Delta\Delta E_{steric}$, $\Delta\Delta E_{elstat}$, $\Delta\Delta E_{orbital}$, and $\Delta\Delta E_{disper}$) is calculated in the same fashion. Positive $\Delta\Delta E$ values indicate corresponding interactions promote *stagger*-type, and *R*-axial chirality TS is more favored. In contrast, negative $\Delta\Delta E$ values indicate *S*-axial chirality TS with *stacking*-type is more stable; $E_{steric}$ is the positive term as the steric repulsion, and $E_{disper}$ is the negative term as the dispersion stabilization. **c** Noncovalent Interactions (NCIs) analysis of transition state for proton transfer (green, dispersion effect; red, steric effect). The gradient isosurfaces (s = 0.3 a.u.) are colored on an BGR scale according to sign($\lambda_2$)$\rho$ over the range $-0.01$ to 0.01 au. **d** Detailed structural information of proton transfer transition states catalyzed by two CPAs. The energies shown in kcal/mol.

---

**Table 2 Calculated free energies of activation for proton transfer transition states for substrates with different R²/R³ substituents.**

| | TS | **R²** | **R³** | exp [a]d.r. | exp [b]$\Delta\Delta G^{\ddagger}$ | calc [b]$\Delta\Delta G^{\ddagger}$ |
|---|---|---|---|---|---|---|
| | **TS3-B** | Ph | Me | 20:1 | 1.8 | 1.7 |
| | | Cy | Me | 4:1 | 0.8 | 1.1 |
| | | Ph | nPr | 23:1 | 1.9 | 3.0 |
| | **TS3-C** | Ph | Me | 12:1 | 1.5 | 1.6 |
| | | Cy | Me | 13:1 | 1.5 | 1.8 |

[a]In **B2** catalysis, the diastereoselective ratio is **3** (major): **4** (minor), and in **C1** catalysis, the diastereoselective ratio is **4** (major): **3** (minor).
[b]The value $\Delta\Delta G^{\ddagger}$ in **TS3-B** is $\Delta G_{stacking\text{-}TS}^{\ddagger} - \Delta G_{stagger\text{-}TS}^{\ddagger}$, and the value $\Delta\Delta G^{\ddagger}$ in **TS3-C** is $\Delta G_{stagger\text{-}TS}^{\ddagger} - \Delta G_{stacking\text{-}TS}^{\ddagger}$. The energy shown in kcal/mol. **Ar** = 3,5-dimethoxyphenyl.

---

**B2** featuring a 1-(4-Me-naphthyl) substitution (over just 1-naphthyl in catalyst **B1**) could improve the diastereoselectivity significantly (2.7:1 to 10:1, see Table 1). For this catalyst, the $\Delta\Delta E_{orbital}$, a high proportion for energy, stabilized for the

unfavorable **TS3-B-SRS**, which can be rationalized by the dihedral angle $\angle$C1-C2-C3-H1 of forming hydrogen bond and allene as shown in Supplementary Fig. 8. The dihedral angle in **TS3-B-SRS** is 85.0°, which is more close to perpendicular than that in **TS3-B-SRR**

**Fig. 6 Derivatizations of the chiral allene products. a** Synthesis of chiral furan derivatives. **b** Derivatizations of the chiral allene products for chemo-diversity-oriented and stereo-diversity-oriented synthesis. **Ar** = 3,5-dimethoxyphenyl.

with 80.2°. However, this orbital interaction cannot overcome the steric and dispersion effect that favor **TS3-B-SRR** as a *stagger*-form in the overall relative energy. The calculated results demonstrate the substrates may dynamically orientate their conformations to interact favorably with various cavities of catalysts under the promotion of inter-/intramolecular interactions.

To confirm validity of the stereo-models in the proton transfer step, we performed additional DFT calculations for understanding substituent effects of $R^2$ and $R^3$ group in **1** as shown in Table 2. The calculated diastereoselectivities are in well agreement with the trend of our experimental substituent effects. In **B2** catalysis, dispersion effects dominate in the stereo-control. When

the cyclohexyl group (Cy) at the R[2] of **1** was introduced and **TS3-B-SRS-Cy** would be stabilized due to the CH-π dispersion effect between cyclohexyl and 4-Me-naphthyl group, which therefore decrease the energy gap for diastereoselectivity. And, n-propyl group (nPr) at R[3] in **1** also emphasize the dispersion effect with 4-Me-naphthyl group of **B2** catalyst, which mainly stabilize the transition state **TS3-B-SRR-nPr** and thus increase the diastereoselectivity. In contrast, the relatively rigid conformation and small cavity for **C1** catalysis make steric repulsions become important. Introducing the relatively larger cyclohexyl group to substrates would increase energy barriers for both the *stacking*-forms and *stagger*-forms of **TS3-C**, resulting slightly increased energy gap for the diastereoselectivity (see Supplementary Table 5 for details). While switching the R[3] group to n-propyl under **C1** catalyst led to very low reactivity by our experiment (see Supplementary Fig. 1), and thus further discussions on the diastereoselectivity is trivial.

**Derivatizations of the chiral allene products for diversity-oriented synthesis**. To demonstrate the applicability of these reactions, we devoted our efforts to exploring the derivatizations of the chiral allene products (Fig. 6). Rearrangement of chiral allene **4a** into tri-substituted furan **7a** was performed under gold (I) catalysis, providing the product in 92% yield. Further alcoholysis of the oxazolone moiety in **7a** provided the β,β-di-aryl substituted amino acid derivative **8a**, which is a type of important pharmacophore in a series of bioactive small molecules. Analogously, gold(I)-catalyzed rearrangement of **3a** afforded the furan derivative **7a** with the same bias on a chiral stationary phase, thus confirming the absolute configuration of **3a** (Fig. 6a). To achieve diversity-oriented synthesis (DOS)[69] from these chiral allene products, diastereoselective reduction of **3a** with L-Selectride followed by in-situ intramolecular transesterification provided the lactone **9a** (>25:1 dr)[70]. Aminolysis of lactone **7a** with nPrNH₂ yielded the allenic alcohol **10a**, which was stereospecifically cyclized in the presence of gold(I) catalyst to give the 2,5-dihydrofuran derivatives **11a**, whose absolute structure was confirmed by X-ray crystallography. Analogously, treatment of chiral allene **4a** with the same three-steps procedure readily provided the diastereomeric chiral 2,5-dihydrofuran **15a** (Fig. 6b). Moreover, derivatizations of the chiral allene products were not limited to stereochemical diversities, but could be extended to skeletal diversities. Electrophilic iodocyclization of lactone **9a** in the presence of NIS provided the tricyclic product **12a**, in which the phenyl group reacted as the nucleophile. On the other hand, treatment of the diastereomeric lactone **13a** with the same electrophilic iodination conditions generated the bridged bicyclic product **16a**, in which the amide group reacted as the nucleophile; the structures of these cyclization products were all well confirmed by X-Ray crystallography.

## Discussion

In summary, we have disclosed an enantioselective and diastereodivergent synthesis of trisubstituted allenes via asymmetric conjugate additions of oxazolones to activated 1,3-enynes under chiral phosphoric acid catalysis, where the axial stereogenicity of the chiral allene products could be well modulated by modifications of the CPA catalysts. The origin of allenic axial diastereodivergence is well elucidated by the *stacking*-form and *stagger*-form in transition states from DFT calculations, in which a Münchnone-type model on activation of oxazolones under Brønsted acid catalysis has also been demonstrated with high model dependency of diastereoselectivities. The stereo-specific and chemo-specific transformations of the diastereomeric chiral allenes into more complex stereoisomers and skeletal isomers

demonstrate the value of these reactions in organic synthesis, especially in the field of DOS.

## Methods

**General procedure for asymmetric synthesis of chiral products 3 and 4**. To a dried 3 ml vial was added **1** (0.15 mmol), **2** (0.1 mmol), CPA catalyst (0.01 mmol), and activated 3 Å molecular sieves (100 mg). The vial was purged with N₂ for 3 times and then followed by adding CCl₄ (0.5 mL). After stirring for 24 h at room temperature, the reaction mixture was quenched by adding K₂CO₃. After filtration, the filtrate was concentrated under vacuum to give a residue, which was purified by flash column chromatography to give the allene products **3 or 4**. (Toluene is an alternative choice of solvent used in these reactions, if the usage of CCl₄ is restricted). Full experimental details and characterization of new compounds can be found in the Supplementary Information.

## Data availability

The authors declare that the data supporting the findings of this study are available within the article and Supplementary Information file, or from the corresponding author upon reasonable request. The X-ray crystallographic coordinates for structures reported in this study have been deposited at the Cambridge Crystallographic Data Centre (CCDC), under deposition numbers CCDC 1971171, 1971946, 1971947, 1971365, and 1971366. These data can be obtained free of charge from The Cambridge Crystallographic Data Centre via www.ccdc.cam.ac.uk/data_request/cif.

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

## Acknowledgements

This work is dedicated to the 70th anniversary of Shanghai Institute of Organic Chemistry and 100th anniversary of the School of Chemistry and Chemical Engineering, Nanjing University. We gratefully acknowledge NSFC (Grant Nos. 21702138, 21890722,

21702109, 11811530637, 21950410519), the Natural Science Foundation of Tianjin Municipality (18JCYBJC21400, 19JCJQJC62300), Tianjin Research Innovation Project for Postgraduate Students (2019YJSB081), ShanghaiTech University start-up funding, and the Fundamental Research Funds for Central Universities [Nankai University (Nos. 63201043, 63203002)] for financial support. The authors thank the support from Analytical Instrumentation Center (# SPST-AIC10112914), SPST, ShanghaiTech University.

## Author contributions

J.W., S.R., and J.X. performed the experiments. S.Z. and Q.P. performed the computational study. N.Y. collected the crystallographic data. Q.P. and X.Y. directed the project and wrote the paper.

## Competing interests

The authors declare no competing interests.
