## [Peer Review File · Nature Communications]

REVIEWER COMMENTS

Reviewer #1 (Remarks to the Author):

The manuscript submitted by X. Yang describes the development of a CPA-catalysed stereodivergent addition of oxazolones to keto-activated 1,3-enynes that leads to trisubstituted allenes. Zhang (CEJ 2008, 8481), Feng (ACIE 2016, 1859), Jørgensen (reference 17) and others have previously described and previously utilised related reactivity of the keto-activated 1,3-enynes. Initial optimisation studies revealed that H8-BINOL-derived- and spirocyclic-CPAs combined with substrates bearing 3,5-dimethoxyphenyl groups give excellent diastereo- and enantioselectivities in CCl₄. Interestingly, the two catalysts give opposite diastereoselectivity and the relative configuration was assigned by X-ray crystallography. The scope for both catalytic systems were examined and the absolute configuration determined by crystal structure analysis for the products from the spirocyclic CPA. DFT calculation supported the mechanistic discussion and rational pathways are proposed. However, the possibility of reversibility of the last protonation step was not described and could be a reason for reduced configurational stability (also the alpha-protonation was described by Zhang). The products could thus interconvert from a kinetic to a thermodynamic epimer by an allene isomerisation and it would be valuable information if this is also the case with the described products. This could also be a reason why variations of the R₃ group were not well tolerated. The products were then derivatised into several interesting scaffolds and the value of the stereodivergent synthesis demonstrated by pathways to different skeletal isomers.

Overall, the concept presented in this manuscript represents an important step in the development of stereodivergent methods and provides very valuable products. It is thus recommended that the possibility of product deprotonation / reversibility and further additional points as listed below are addressed in major revisions before acceptance for publication.

- 1) Table 1 contains Ar groups for substrates and catalysts: these should be differentiated.
- 2) ≠ should be replaced with the ‡ symbol for all transition states.
- 3) 3,5-dimethoxyphenyl not 3,5-dimethoxy(l)phenyl
- 4) Provide alternatives to CCl₄ that will ensure unrestricted adoption of the method.
- 5) Configurational stability: a) Which is the thermodynamic product and b) what are the limits for acidic and basic conditions that maintain the configuration of the products.
- 6) Show other enantiomer of 3m for consistency
- 7) Provide the Flack parameters for all non-racemic structures in the SI.
- 8) Provide the assignment of the minor stereoisomers by correlation with the other catalytic system.
- 9) Zhang and Feng should be cited.

Reviewer #2 (Remarks to the Author):

The authors describe enantioselective and diastereodivergent synthesis of trisubstituted allenes via asymmetric conjugate additions of activated 1,3-enynes by oxazolones enabled by chiral phosphoric acid catalysis. The rarely reported stereodivergent synthesis of axial stereogenicity using a single type of catalyst backbone has been achieved. The diastereodivergence has also computationally addressed and allowed us to gain deep insights into the mechanistic details. This referee recommends this paper to be accepted for publication in Nature Communications after following corrections regarding computational part:

- (1) As described by authors (line 196), B2 catalyst fails to determine stereoselectivity in the Michael addition step but control (S,R,R)-configuration in the proton transfer step whereas C1

catalyst can control (S,R)-configuration followed by (S)-axial chirality in both steps, respectively. Although the origin of chirality control in the proton transfer step was pointed out in detail, that in the Michael addition step was NOT explained enough. The similarity and difference in between B2 and C1 catalysis particularly regarding stereocontrol ability should be clarified in this paper. In Figure S3, distortion-interaction analysis of TS2-B indicates the substrate distortion destabilizes TS2-B-SR, decreasing the energy difference with TS2-B-RS. In my opinion, the Ph group (R2) at the beta position of 1a induces sterically repulsive interactions with 3,3'-substituent of B2 catalyst as well as the Et group of 2a. Did authors explore more stable conformational isomer of TS2-B-SR model regarding the Et group rotation in 2a (Enough space seems to exist around the Et group)? If the Michael adduct in racemic form could be prepared, treatment of it under the same reaction condition gives us a chance to experimentally confirm reversibility of the Michael addition step in the B2 catalysis.

(2) For line 213 and Figure S4, authors describe that smaller distortion energies of substrate fragment in stacking TS3-SRS are originated from intramolecular interactions such as π - π interaction. It seems to be overestimated and exaggerated. Focusing on the direction of the Ph group at the R1 position of 1, the resonance stabilization between the Ph group (R1) and the allenic moiety in 1a also decreases the distortion energy of substrate fragment in TS3-SRS.

(3) For line 230 and Fig. 5, authors describe that 1-(4-Me-naphthyl) substitution emphasizes the dispersion effect around the dotted circle region (Fig. 5a) to stabilize TS3-B-SRR. In my opinion, the dominant dispersion effect in TS3-SRS would be also originated from the 3,5-(MeO)₂C₆H₃ group of 2a. NCI plot should be conducted to clarify the dominant dispersion effect.

(4) To confirm validity of TS model, the experimentally observed substituent effects should be rationally explained based on the proposed TS model. For example, the notable difference in between B2 and C1 catalysis were exhibited in the substituted effect on the R3 group of 1. In the C1 catalysis, the variations of the R3 groups of 1 were not well tolerated, decreasing diastereoselectivity (the corresponding data should be added). Did authors explore not only n-alkyl group but also i-Pr and/or t-Bu? The n-alkyl group at the R3 position would emphasize the CH- π interaction with 1-Naph group of C1 catalyst, which does not exist in TS3-C-SRS, stabilizing TS3-C-SRR and decreasing diastereoselectivity. Is there a chance to achieve high diastereoselectivity using 1 with the sterically demanding alkyl groups at the R3 position by destabilizing TS3-C-SRR through the steric repulsion? In addition, both R1 and R2 groups have a greater impact on diastereoselectivity of B2 catalysis than C1 catalysis. Such experimentally observed substituent effects should be discussed based on TS3 models.

(5) For line 278, "Discussion" should be "Conclusion".

1. Reviewer #1

Comments: *The manuscript submitted by X. Yang describes the development of a CPA-catalysed stereodivergent addition of oxazolones to keto-activated 1,3-enynes that leads to trisubstituted allenes. Zhang (CEJ 2008, 8481), Feng (ACIE 2016, 1859), Jørgensen (reference 17) and others have previously described and previously utilised related reactivity of the keto-activated 1,3-enynes. Initial optimisation studies revealed that H8-BINOL-derived- and spirocyclic-CPAs combined with substrates bearing 3,5-dimethoxyphenyl groups give excellent diastereo- and enantioselectivities in CCl₄. Interestingly, the two catalysts give opposite diastereoselectivity and the relative configuration was assigned by X-ray crystallography. The scope for both catalytic systems were examined and the absolute configuration determined by crystal structure analysis for the products from the spirocyclic CPA. DFT calculation supported the mechanistic discussion and rational pathways are proposed. However, the possibility of reversibility of the last protonation step was not described and could be a reason for reduced configurational stability (also the alpha-protonation was described by Zhang). The products could thus interconvert from a kinetic to a thermodynamic epimer by an allene isomerisation and it would be valuable information if this is also the case with the described products. This could also be a reason why variations of the R³ group were not well tolerated. The products were then derivatised into several interesting scaffolds and the value of the stereodivergent synthesis demonstrated by pathways to different skeletal isomers.*

Overall, the concept presented in this manuscript represents an important step in the development of stereodivergent methods and provides very valuable products. It is thus recommended that the possibility of product deprotonation / reversibility and further additional points as listed below are addressed in major revisions before acceptance for publication.

Response: We thank the reviewer for careful reading and constructive comments. Some control experiments on the stabilities of chiral allene products under various conditions were performed (see Fig. S2 in Supporting Information). Treatment of allene **3a** with CPA (*S*)-**B2** and allene **4a** with CPA (*R*)-**C1** didn't lead to the isomerization between these two products. Additionally, monitoring the reaction by ¹H NMR also indicated the diastereomeric ratio did not change during the reaction progress, which indicated the conversion from a kinetic epimer to a thermodynamic epimer under the reaction conditions is probably not likely.

However, interestingly, treatment of the allene products with Et₃N (1.0 equiv. or as solvent) led to the significant isomerization between these two products, which suggested the product deprotonation/reversibility is possible under basic conditions.

Comments:

1) *Table 1 contains Ar groups for substrates and catalysts: these should be differentiated.*

Response: The "Ar" in the catalysts has been revised to "R".

Comments:

2) \neq should be replaced with the \ddagger symbol for all transition states.

Response: We thank the referee for pointing this out and we have fixed them.

Comments:

3) 3,5-dimethoxyphenyl not 3,5-dimethoxy(l)phenyl

Response: This typo has been revised accordingly.

Comments:

3) Provide alternatives to CCl_4 that will ensure unrestricted adoption of the method.

Response: Toluene is an alternative choice of solvent used in these reactions, in which the chiral allene **3a** was obtained in 99% yield, 12:1 dr (**3a:4a**) with 91% ee in the presence of CPA **B2**, while diastereomeric chiral allene **4a** was generated in 80% yield, 9:1 dr (**4a:3a**) with 94% ee under the catalysis of CPA **C1**. This statement has been added in the "Method" section of the manuscript.

Comments:

4) *Configurational stability: a) Which is the thermodynamic product and b) what are the limits for acidic and basic conditions that maintain the configuration of the products.*

Response: Some control experiments on the stabilities of chiral allene products under various conditions were performed (see Fig. S2 in Supporting Information). Treatment of allene **3a** with CPA (*S*)-**B2** and allene **4a** with CPA (*R*)-**C1** didn't lead to the isomerization between these two products. Additionally, monitoring the reaction by ^1H NMR also indicated the diastereomeric ratio did not change during the reaction progress, which indicated the conversion from a kinetic epimer to a thermodynamic epimer under the reaction conditions is probably not likely.

The allene products are generally stable under thermal conditions (60 °C) and mild acidic conditions (50 mol% CPA or 1 equiv. HOAc in toluene), and almost no isomerization between the two diastereomers was observed. However, treatment of the allene products with strong acidic conditions (e.g. 1.0 equiv. TFA in toluene) provided a complex mixture, which included the oxazolone hydrolysis products, along with the rearrangement furan derivative **7a**.

The allene products were vulnerable under basic conditions. Treatment of the allene products with Et_3N (1.0 equiv.) led to the significant isomerization between these two products. Furthermore, treatment of allene **3a** and **4a** with Et_3N (as solvent) provided a mixture of these two products with approximate the same ratio (**3a:4a** ~ 0.4:0.6), which suggested the allene **4a** was the relatively thermodynamically stable diastereomer under basic conditions. Treatment with the allene products with NaOH led to the elimination of oxazolone and regeneration of the α -alkynyl enone **1a** as the major product.

Comments:

5) *Show other enantiomer of 3m for consistency.*

Response: The X-ray structure of the other enantiomer of **3m** has been showed for consistency.

Comment:

6) *Provide the Flack parameters for all non-racemic structures in the SI.*

Response: The single crystal data (including the Flack parameters) for all X-ray structures have been provided in Supporting Information.

Comment:

7) *Provide the assignment of the minor stereoisomers by correlation with the other catalytic system.*

Response: The absolute configurations of the minor stereoisomers was assigned by comparison of their HPLC traces (see HPLC spectra of allene products **3w** and **4w**) with the other catalytic system, which suggested the minor stereoisomers in each catalytic system have the same two point chiralities with the major diastereomers, but with different axial chirality.

Comments:

8) *Zhang and Feng should be cited.*

Response: These two references have been cited as ref 11 and 19, respectively.

2. Reviewer #2

Comments : *The authors describe enantioselective and diastereodivergent synthesis of trisubstituted allenes via asymmetric conjugate additions of activated 1,3-enynes by oxazolones enabled by chiral phosphoric acid catalysis. The rarely reported stereodivergent synthesis of axial stereogenicity using a single type of catalyst backbone has been achieved. The diastereodivergence has also computationally addressed and allowed us to gain deep insights into the mechanistic details. This referee recommends this paper to be accepted for publication in Nature Communications after following corrections regarding computational part:*

Response: We thank the anonymous reviewers for careful reading and constructive comments. We have carefully addressed the referees' concerns. You will find a detailed list below.

Comments:

(1) *As described by authors (line 196), B2 catalyst fails to determine stereoselectivity in the Michael addition step but control (S,R,R)-configuration in the proton transfer step whereas C1 catalyst can control (S,R)-configuration followed by (S)-axial chirality in both steps, respectively. Although the origin of chirality control in the proton transfer step was pointed out in detail, that in*

the Michael addition step was NOT explained enough. The similarity and difference in between B2 and C1 catalysis particularly regarding stereocontrol ability should be clarified in this paper. In Figure S3, distortion-interaction analysis of TS2-B indicates the substrate distortion destabilizes TS2-B-SR, decreasing the energy difference with TS2-B-RS. In my opinion, the Ph group (R^2) at the beta position of 1a induces sterically repulsive interactions with 3,3'-substituent of B2 catalyst as well as the Et group of 2a. Did authors explore more stable conformational isomer of TS2-B-SR model regarding the Et group rotation in 2a (Enough space seems to exist around the Et group)? If the Michael adduct in racemic form could be prepared, treatment of it under the same reaction condition gives us a chance to experimentally confirm reversibility of the Michael addition step in the B2 catalysis.

Response: From computational point of view, the more stable conformation of TS2-B-SR had been located by rotating the ethyl group that is 1.8 kcal/mol more stable than our original TS2-B-SR. Together with the intermediate INT3-B-SR, the Fig. 4d and Fig.S3b has been revised. And the distortion interaction analysis has been updated in Fig. S5b. After carefully checking the calculated stereo-structures, we agree the referee's comment that the phenyl group (R^2) at the beta position of 1a induces sterically repulsive interactions with the ethyl group of 2a. And the corresponding revisions had been revised and added in main text and SI that were highlighted in yellow. Our main conclusions will not be changed by the above revision.

Fig. 4d) Control experiments and DFT calculations of the reaction mechanism. The energy shown in kcal/mol.

Fig. S5. (b) Distortion-interaction analysis of Michael-Addition TS performed at M06-2X/6-311++G(d,p) level of theory in Gaussian 09.

No Michael addition intermediate could be detected or isolated in these reactions. It is very challenging to prepare the α -alkynyl substituted ketones, because they readily undergo the isomerization to give the allenes under acid or basic conditions. For example, substrate **5a** was prepared without purification via silica gel column chromatography, because purification of these compounds with silica gel column chromatography would also lead to the isomerization to allenes (for details see SI).

Comments:

(2) For line 213 and Figure S4, authors describe that smaller distortion energies of substrate fragment in stacking TS3-SRS are originated from intramolecular interactions such as π - π interaction. It seems to be overestimated and exaggerated. Focusing on the direction of the Ph group at the R^1 position of **1**, the resonance stabilization between the Ph group (R^1) and the allenic moiety in **1a** also decreases the distortion energy of substrate fragment in TS3-SRS.

Response: We agree that the resonance stabilization between the phenyl group (R^1) and the allenic moiety in **1a** would decrease the distortion energy of substrate fragment in TS3-SRS. The torsional angle $\angle C4-C3-C5-C6$ were shown to depict the coplanarity of Ph group (R^1) and the allenic moiety in **1a**. The explanation for this part is added in the caption of Fig. S6. And the revisions in main text were highlighted.

Fig. S6 Distortion-interaction analysis of TSs for proton transfer performed at M06-2X/6-311++G(d,p) level of theory in Gaussian 09. For both the catalysts (*R*)-SPINOL-C1 and (*S*)-H8-BINOL-B2, the distortion-interaction analysis reveals that the interactions between catalyst fragment and substrate fragment are more favorable in *stagger*-type TSs. And the distortion energies of substrate fragment are lower in *stacking*-type TSs, which indicate that the substrate intermediate has strong intramolecular interactions. In addition, focusing on the direction of the phenyl group at the R¹ position of α -alkynyl enones 1a, the dihedral angles \angle C4-C3-C5-C6 of Phenyl group (R¹) and the allenic moiety in *stacking*-type TSs, which are closer to planar than that in *stagger*-type TSs. The coplanarity of the dihedral angle results in the resonance stabilization between the phenyl group (R¹) and the allenic moiety, which also decreases the distortion energy of substrate fragment in *stacking*-type TSs.

Comments:

(3) For line 230 and Fig. 5, authors describe that 1-(4-Me-naphthyl) substitution emphasizes the dispersion effect around the dotted circle region (Fig. 5a) to stabilize TS3-B-SRR. In my opinion, the dominant dispersion effect in TS3-SRS would be also originated from the 3,5-(MeO)₂C₆H₃ group of 2a. NCI plot should be conducted to clarify the dominant dispersion effect.

Response: The NCI plots for TS3-B-SRR and TS3-C-SRS have been illustrated in Fig. S7. For the dotted circle region in main text of Fig. 5a, we can find the important dispersion effect colored in green. And the dispersion effect originated from the 3,5-(MeO)₂C₆H₃ group of 2a is shown in TS3-SRS that is π - π interaction. And we thank the referee for this kind suggestion of weak interactions. NCI plots were added and the relevant was in Fig. 5c. And computational details were updated for NCI plots visualization.

Fig. S7 NCIs analysis of transition state for proton transfer (green, dispersion effect; red, steric effect). There is weakening π - π stacking between allenic moiety (include Phenyl group) of 1a and oxazolone moiety (include Ar group) of 2a in *stacking*-type TSs. And, there are clear dispersion

effect between the naphthalyl or 1-(4-Me)-naphthyl group of the catalysts and acetyl of **1a**, as well as between spiro ring or binaphthyl moiety of catalyst and 3,5-(MeO)₂C₆H₃ group of **2a** in *stagger*-type TSs. Therefore, the intramolecular interactions mainly exist in *stacking*-type TSs, and intermolecular interactions mainly occur in *stagger*-type TSs.

Comments:

(4) To confirm validity of TS model, the experimentally observed substituent effects should be rationally explained based on the proposed TS model. For example, the notable difference in between **B2** and **C1** catalysis were exhibited in the substituted effect on the R³ group of **1**. In the **C1** catalysis, the variations of the R³ groups of **1** were not well tolerated, decreasing diastereoselectivity (the corresponding data should be added). Did authors explore not only *n*-alkyl group but also *i*-Pr and/or *t*-Bu? The *n*-alkyl group at the R³ position would emphasize the CH- π interaction with 1-Naph group of **C1** catalyst, which does not exist in TS3-C-SRS, stabilizing TS3-C-SRR and decreasing diastereoselectivity. Is there a chance to achieve high diastereoselectivity using **1** with the sterically demanding alkyl groups at the R³ position by destabilizing TS3-C-SRR through the steric repulsion? In addition, both R¹ and R² groups have a greater impact on diastereoselectivity of **B2** catalysis than **C1** catalysis. Such experimentally observed substituent effects should be discussed based on TS3 models.

Response: The results for the scope of variants of the R³ group of **1** under the catalysis of (*R*)-**C1** catalyst were provided in Fig. S1 in Supporting Information, which generated the chiral allene products **4** in low yield with poor diastereoselectivities, although the enantioselectivities of generated allene **4** were still good.

Switching of R³ group from primary alkyl groups to secondary alkyl group was also studied (with a substrate (**S1v**) possessing an isopropyl group, see Fig. S1 in Supporting Information), however, which showed no reactivities under the catalysis of (*R*)-**B2** or (*R*)-**C1** catalyst at ambient temperature. Raising the reaction temperature to 55 °C provided the desired allene product in poor yield, low diastereoselectivities and enantioselectivities and no diastereodivergence was observed in this reaction.

We also performed additional DFT calculations for understanding substituent effects of R² and R³ group in **1** to confirm validity of the stereo-models in the proton transfer step. Calculations for switching the R³ group under the catalysis of (*R*)-**C1** were ruled out due to very low reactivity in our experiment (Fig. S1 in SI). Computational results are in well agreement with the trend of our experimental substituent effects, see Table 2 and S5 for details. And related discussions were highlighted in main text. A more stable TS3-C-SRR by rotating the conformation of -OMe was located and the results were updated in Fig. 4, 5 and S6, accordingly.

Comments:

(5) For line 278, "Discussion" should be "Conclusion".

Response: It has been revised accordingly.

REVIEWERS' COMMENTS

Reviewer #1 (Remarks to the Author):

In their revised manuscript, the author have appropriately addresses the points raised by the reviewers. In particular the reversibility of the last step was carefully examined, providing the crucial confirmation about kinetic and thermodynamic product formation. This reviewer therefore recommends to accept this revised manuscript for publication.

Reviewer #2 (Remarks to the Author):

The manuscript has been improved properly. The additional NCI analysis makes us easily find the important dispersion effect. The revised discussion about Distortion-Interaction analysis improves readability. The DFT computations regarding the substituent effects and experimental results seem to be reasonably connected, confirming validity of TS model. This is now an excellent paper containing interesting results which merit publication in Nature Communications.